Ectopic expression of HIV-1 Tat modifies gene expression in cultured B cells: implications for the development of B-cell lymphomas in HIV-1-infected patients

Valyaeva Anna A. 1 2 3
Tikhomirova Maria A. 1 4
http://orcid.org/0000-0002-0098-827X Potashnikova Daria M. 3
Bogomazova Alexandra N. 5 6
Snigiryova Galina P. 7
Penin Aleksey A. 8
Logacheva Maria D. 2 9
Arifulin Eugene A. 2
Shmakova Anna A. 4 10
Germini Diego 10
Kachalova Anastasia I. 3
http://orcid.org/0000-0003-3294-7146 Saidova Aleena A. 3 11
Zharikova Anastasia A. 1 2
Musinova Yana R. 2 4
Mironov Andrey A. 1 8
http://orcid.org/0000-0003-3101-7043 Vassetzky Yegor S. 4 10 yegor.vassetzky@cnrs.fr
http://orcid.org/0000-0003-1687-1321 Sheval Eugene V. 1 2 3 sheval_e@belozersky.msu.ru
1 School of Bioengineering and Bioinformatics, Lomonosov Moscow State University , Moscow , Russia
2 Belozersky Institute of Physico-Chemical Biology, Lomonosov Moscow State University , Moscow , Russia
3 Department of Cell Biology and Histology, School of Biology, Lomonosov Moscow State University , Moscow , Russia
4 Koltzov Institute of Developmental Biology , Moscow , Russia
5 Federal Research and Clinical Center of Physical-Chemical Medicine , Moscow , Russia
6 Center for Precision Genome Editing and Genetic Technologies for Biomedicine, Federal Research and Clinical Center of Physical-Chemical Medicine of Federal Medical Biological Agency , Moscow , Russia
7 Burdenko National Medical Research Center of Neurosurgery , Moscow , Russia
8 Institute for Information Transmission Problems , Moscow , Russia
9 Skolkovo Institute of Science and Technology , Moscow , Russia
10 UMR9018 (CNRS – Institut Gustave Roussy – Université Paris Saclay), Centre National de Recherche Scientifique , Villejuif, France , France
11 Center for Precision Genome Editing and Genetic Technologies for Biomedicine, Engelhardt Institute of Molecular Biology , Moscow , Russia
Tandon Ravi
Electronic publication date: 2022 Oct 18
Publication date: 2022
Volume: 10
Electronic Location ID: e13986
Received 2022 May 12; Accepted 2022 Aug 11
Copyright: © 2022 Valyaeva et al.
Copyright year: 2022
Copyright holder: Valyaeva et al.
License: This is an open access article distributed under the terms of the Creative Commons Attribution License, which permits unrestricted use, distribution, reproduction and adaptation in any medium and for any purpose provided that it is properly attributed. For attribution, the original author(s), title, publication source (PeerJ) and either DOI or URL of the article must be cited.
License URL: https://creativecommons.org/licenses/by/4.0/

Keywords: HIV-1 Tat, B cells, Virus-cell interactions, Gene expression, RNA-seq

Funding: Russian Science Foundation 21-74-20134 Russian Foundation for Basic Research 20-34-90156 This research was funded by the Russian Science Foundation (grant 21-74-20134 to Eugene V. Sheval) and the Russian Foundation for Basic Research (PhD student grant 20-34-90156 to Maria A. Tikhomirova). The funders had no role in study design, data collection and analysis, decision to publish, or preparation of the manuscript.

==============================
An increased frequency of B-cell lymphomas is observed in human immunodeficiency virus-1 (HIV-1)-infected patients, although HIV-1 does not infect B cells. Development of B-cell lymphomas may be potentially due to the action of the HIV-1 Tat protein, which is actively released from HIV-1-infected cells, on uninfected B cells. The exact mechanism of Tat-induced B-cell lymphomagenesis has not yet been precisely identified. Here, we ectopically expressed either Tat or its TatC22G mutant devoid of transactivation activity in the RPMI 8866 lymphoblastoid B cell line and performed a genome-wide analysis of host gene expression. Stable expression of both Tat and TatC22G led to substantial modifications of the host transcriptome, including pronounced changes in antiviral response and cell cycle pathways. We did not find any strong action of Tat on cell proliferation, but during prolonged culturing, Tat-expressing cells were displaced by non-expressing cells, indicating that Tat expression slightly inhibited cell growth. We also found an increased frequency of chromosome aberrations in cells expressing Tat. Thus, Tat can modify gene expression in cultured B cells, leading to subtle modifications in cellular growth and chromosome instability, which could promote lymphomagenesis over time.

Introduction

In the second part of the 20th century, human immunodeficiency virus-1 (HIV-1) has rapidly spread throughout the world and caused high mortality due to its high evolution rate. HIV-1 preferentially infects CD4+ T cells, macrophages, and microglial cells, leading to a damaged immune system and the development of acquired immunodeficiency syndrome (AIDS). Combined antiretroviral therapy (cART) stops the virus from making copies of itself in the body and may slow down the development of AIDS (Autran et al., 1997; HIV-CAUSAL Collaboration et al., 2010). However, even after the introduction of cART, individuals infected with HIV-1 are at significantly higher risk of developing non-AIDS-related comorbidities, including the development of neurocognitive disorders (Mateen et al., 2012; Marino et al., 2020), cardiovascular diseases (Wang et al., 2015; Jiang et al., 2018), adipose tissue senescence (Gorwood et al., 2020), and cancer (Shmakova, Germini & Vassetzky, 2020).

Despite the improved control of HIV-1 infection achieved by cART, B-cell lymphomas are still elevated in HIV-1-infected patients and are the most frequent cause of death in these patients (Noy, 2020; Shmakova, Germini & Vassetzky, 2020; Hübel, 2020). However, even more surprising is the fact that HIV-1-infected patients have an increased incidence of specific B-cell lymphomas, namely Burkitt lymphoma and diffuse large B-cell lymphoma (Gloghini, Dolcetti & Carbone, 2013; Besson et al., 2017; Atallah-Yunes, Murphy & Noy, 2020). Only a few articles report that B cells can be infected with HIV-1 (Fritsch et al., 1998; Lazzi et al., 2002; Katano et al., 2007), and it seems that this is an extremely rare/artifactual event. One of the most likely mechanisms of the development of HIV-1-associated B-cell lymphomas in HIV-1-infected patients may be an action of viral proteins on uninfected B cells (Dolcetti et al., 2016).

HIV-1 produces a small nuclear transcriptional activator protein known as transactivator of transcription (Tat) which regulates viral transcription (Ali et al., 2021). In the absence of Tat, HIV-1 proviral promoter is incompetent for elongation: shortly after transcription initiation, RNA Pol II is stalled due to the presence of inactive form of positive transcription elongation factor b (P-TEFb) composed of CDK9, cyclin T1 or T2, and inhibitory 7SK small nuclear ribonucleoprotein complex (containing 7SK RNA and HEXIM1) (Kao et al., 1987; Sedore et al., 2007; D’Orso & Frankel, 2010). Tat is able to relieve this repressed state by binding to the TAR-like sequence in 7SK snRNA and displacing HEXIM1 from cyclin T1, which disrupts the 7SK-P-TEFb negative transcriptional regulatory complex and releases active P-TEFb (Barboric et al., 2007; Sedore et al., 2007; Muniz et al., 2010; Pham et al., 2018). This ability of Tat depends on Tat C22 residue within the activation domain (Barboric et al., 2007). Tat then recruits the active P-TEFb complex (consisting of CDK9, cyclin T1 or T2) and other transcriptional coactivators to the TAR RNA element at the 5′ end of stalled nascent HIV-1 transcripts to relieve stalled RNA Pol II (Nekhai & Jeang, 2006; He et al., 2010). Simultaneously, HIV-1 Tat modulates cellular processes by interacting with different cellular structures, particularly nuclear components (Musinova et al., 2016; Ali et al., 2021).

Expression of HIV-1 Tat alone in mice leads to development of different neoplasms, including lymphomas (Vogel et al., 1988, 1991; Corallini et al., 1993; Altavilla et al., 1999; Kundu et al., 1999), suggesting that Tat protein participates in oncogenesis in HIV-1-infected patients. Tat is actively released from HIV-1-infected cells (Ensoli et al., 1990; Nath, 2015) and is detectable in the serum of HIV-1-infected individuals (Westendorp et al., 1995; Xiao et al., 2000; Poggi et al., 2004; Germini et al., 2017). Exogenous HIV-1 Tat can enter uninfected cells, and in particular, HIV-1 Tat is present within tumor cells of HIV-1-associated B-cell lymphomas (Lazzi et al., 2002; Alves de Souza Rios et al., 2021). Some other HIV-1 proteins might also affect cells not infectable by HIV-1, e.g., HIV-1 Nef, which can be secreted in a form of extracellular vesicles and released into circulation (Raymond et al., 2011; Pushkarsky et al., 2022).

The mechanisms of Tat-dependent lymphomagenesis in HIV-1-infected patients have been described only partially. Incubation of B cells from healthy donors with recombinant HIV-1 Tat ex vivo led to the convergence of chromosomal loci that are usually involved in t(8;14) translocation, which is common in Burkitt lymphoma (Germini et al., 2017). These data indicate that exogenous viral proteins can induce global rearrangement of nuclear organization and that these changes can promote lymphomagenesis. Additionally, HIV-1 Tat can modify the chromatin organization and gene expression of host cells, particularly T cells (Marban et al., 2011; Reeder et al., 2015) and macrophages (Carvallo et al., 2017), and it seems that Tat can induce a chain of similar events in B cells. Indeed, Tat can affect the expression of several genes in B cells, including AICDA, a gene that encodes the activation-induced cytidine deaminase that participates in immunoglobulin gene maturation (Sall et al., 2019; Akbay et al., 2021). Additionally, HIV-1 Tat enhances c-MYC transcription by binding to the c-MYC promoter, which can contribute to a more aggressive lymphoma phenotype (Lazzi et al., 2002; Alves de Souza Rios et al., 2021). Thus, HIV-1 Tat present in blood may affect gene expression in B cells, and these changes can promote lymphomagenesis. However, the effect of Tat on B cells has never been studied on a genome-wide level. Here, we ectopically expressed HIV-1 Tat in the lymphoblastoid B cell line (RPMI 8866) and analyzed host gene expression by RNA-seq. We found that the expression of HIV-1 Tat led to substantial modifications of gene expression and induced cellular antiviral reactions. Ectopic Tat expression also resulted in modification of cellular proliferation and genome stability, thus promoting changes that could facilitate lymphomagenesis.

Materials and Methods

Cell culture

RPMI 8866 cells (Sigma, St. Louis, MI, USA) were grown at 37 °C in RPMI 1640-Gluta-Max medium (Gibco, Waltham, MA, USA) supplemented with 10% fetal bovine serum (HyClone, Logan, UT, USA), sodium pyruvate (PanEco, Singapore), and an antibiotic and antimycotic solution (Gibco, Waltham, MA, USA).

HeLa cells with integrated LTR-TurboRFP (Kurnaeva et al., 2022) were grown in Dulbecco’s modified Eagle’s medium supplemented with alanyl-glutamine (Paneco, Singapore), 10% fetal calf serum (HyClone, Logan, UT, USA) and an antibiotic and antimycotic solution (Gibco, Waltham, MA, USA). The transactivation assay based on fast-maturing TurboRFP protein was described elsewhere (Kurnaeva et al., 2022). The expression of EGFP and TurboRFP was analyzed using a FACS Aria SORP instrument (BD Biosciences, San Jose, CA, USA).

Plasmids and cell lines

The pGST-Tat 1 86R plasmid was obtained through the NIH AIDS Reagent Program, Division of AIDS, NIAID, from Dr. Andrew Rice (Herrmann & Rice, 1993, 1995; Rhim et al., 1994).

Plasmids for EGFP, Tat-EGFP and TatC22G-EGFP expression and lentiviral particles were constructed by Evrogen (Moscow, Russia). EGFP-, Tat-EGFP- or TatC22G-EGFP-expressing cells were collected using a FACSAria SORP cell sorter (BD Biosciences, San Jose, CA, USA). The excitation wavelength for EGFP was 488 nm, and the emission was detected by a 505LP and 515/20BP set of filters. Sorting was performed with an 85 μm nozzle and the corresponding custom pressure parameters. The sorted cells were grown, then frozen in the complete medium in the presence of DMSO, and stored in liquid nitrogen. To achieve better reproducibility of experiments, cells were used for no more than one month after thawing (excluding the experiments on long-term culture).

Cell lysate preparation, SDS-PAGE and western blotting

Cells were collected by centrifugation for 10 min at 800 g. Cell pellets were washed with PBS and resuspended in NETN buffer (150 mM NaCl, 1 mM EDTA, 50 mM Tris pH 7.5, 0.5% NP 40, protease inhibitor cocktail), sonicated, incubated on ice for 30 min and centrifuged at 4 °C at 12,000 g for 10 min. Protein quantification was performed using the Pierce™ BCA Protein Assay Kit (Thermo Scientific, Waltham, MA, USA) on a NanoDrop 2000C (Thermo Scientific, Waltham, MA, USA). After measuring the concentration, cell lysates were supplemented with Laemmli buffer and 0.1 M DTT and then heated at 95 °C for 10 min.

Protein samples (20 µg) and prestained molecular weight markers (PageRuler™ Prestained Plus Protein Ladder; Thermo Scientific, Waltham, MA, USA) were resolved on 15-well precast SDS–PAGE gels (4–12%) (NuPage) in MOPS Running Buffer (NuPage). Proteins were transferred onto a 0.45 μm PVDF membrane (GE Healthcare, Chicago, IL, USA) in transfer buffer (0.025 M Tris, 0.192 M glycine, 20% ethanol) at 90 V at 4 °C for 2 h. Nonspecific binding was blocked in 5% nonfat dried milk in Tris-buffered saline and 0.1% Tween-20 (TBST) at room temperature for 1 h.

Proteins were probed at 4 °C overnight with the following primary antibodies: anti-Tat (1:200, cat. #sc-65912; Santa Cruz, CA, USA), anti-GFP (1:1,000, Roche, cat. #11814460001), anti-β-actin (1:1,000, control of protein load, cat. #sc-81178; Santa Cruz, CA, USA). The membranes were washed with TBST and incubated with goat anti-mouse IgG-HRP (cat. # sc-2005; Santa Cruz, CA, USA) secondary antibodies at a 1:2,000 dilution at room temperature for 1.5 h, followed by washing in TBST. Proteins were visualized using Immobilon Western Chemiluminescent HRP Substrate (Millipore, Burlington, MA, USA) and ImageQuant LAS 4000 mini (GE Healthcare, Chicago, IL, USA) for western blotting imaging and analysis.

RNA extraction and sequencing

Cells were collected and stored in RNALater (Qiagen, Hilden, Germany). Total RNA was extracted using a RNeasy Mini RNA isolation kit (Qiagen, Hilden, Germany) following the Qiagen protocol, with the following modifications: (1) lysis time was increased up to 40 min; (2) on-column DNase I treatment was performed. RNA sample quality was assessed using a capillary electrophoresis Bioanalyzer 2100 (Agilent, Santa Clara, CA, USA), and all samples had RIN >8. The cDNA libraries were constructed using the NEBNext Ultra II Directional RNA Library Prep Kit for Illumina (NEB) following the manufacturer’s recommendations. RNA was fragmented for 5 min. Thereafter, the constructed libraries were sequenced on an Illumina HiSeq 2000 with a single-end 51 bp read length. Basecalling was performed using bcl2fastq v2.17.1.14.

RNA-seq data processing and analysis

Read quality control was performed using FastQC (version 0.11.7) (http://www.bioinformatics.babraham.ac.uk/projects/fastqc/). Due to the good quality of the reads, no filtering or adapter removal was performed. The reads were then aligned to the human genome assembly GRCh38.p10 using HISAT2 (version 2.0.5) (Kim, Langmead & Salzberg, 2015). Read counting was performed in strand-specific mode by the htseq-count script from Python library HTSeq (version 0.12.4) (Anders, Pyl & Huber, 2015) using GENCODE v26 gene annotation (ALL). Genes with no counts for all samples, as well as highly expressed ribosomal genes, were filtered out, resulting in the expression set of 32,120 genes across 12 samples.

Principal component analysis (PCA) was performed on the rlog-transformed count data, and the first two principal components were extracted with the corresponding percentage of explained variance. Differential expression analysis was performed with the R package DESeq2 (version 1.30.1) (Love, Huber & Anders, 2014). We declared the gene to be differentially expressed if padj (p value adjusted by the Benjamini-Hochberg procedure) was smaller than 0.05 and the fold change was larger than 1.5 in any direction (Table S1). The statistical power for protein-coding genes (with median 80 aligned reads) was calculated with R package RNASeqPower (version 1.30.0) and was 0.91. Each group of samples consisted of three biological replicates.

Overrepresentation analysis (ORA) and gene set enrichment analysis (GSEA) (Subramanian et al., 2005) were performed with the R package clusterProfiler (version 3.18.1) (Yu et al., 2012). GSEA was performed on a preranked list of genes ordered by the stat column of DESeq2 results. The KEGG (release 99) (Kanehisa & Goto, 2000) and GO biological process (Ashburner et al., 2000; The Gene Ontology Consortium, 2017) databases were used as the sources of gene sets for ORA and GSEA. GO annotation was obtained from the R package org.Hs.eg.db (version 3.12.0). An adjusted p value cutoff of 0.05 was used to select statistically significant categories. REVIGO (Supek et al., 2011) with a cutoff parameter of 0.4 was used to remove redundant GO terms.

To assess the possible additional activation of EBV genes due to Tat protein expression in RPMITat cells, the same pipeline for differential expression analyses was used. The EBV gene annotation file in GTF format was obtained from GenBank file NC_007605.1 using a custom Python script. Raw reads were aligned to combined human and viral genomes. The EBV GTF annotation file was used to obtain counts for viral genes, and previously obtained human gene counts were used to estimate size factors for all samples. Subsequent differential expression analysis was performed for viral genes.

RNA extraction and qRT-PCR assays

Total RNA from RPMI 8866 cells was isolated using the RNeasy Mini Kit (Qiagen, Hilden, Germany). The RNA concentration was measured with a NanoPhotometer (Implen, Westlake Village, CA, USA). Reverse transcription was performed with an iScript Advanced cDNA Synthesis Kit (BioRad, Hercules, CA, USA) according to the manufacturer’s instructions, and qPCR was performed in technical triplicates using a SYBR Green kit (Syntol) in a CFX96 Real-Time PCR Detection System (BioRad, Hercules, CA, USA). Melting curve analyses were performed to verify the amplification specificity. Experiments were performed in biological triplicates, and error bars represent the SEM as indicated in all figure legends. The HPRT, YWHAZ and UBC2 genes were used as references. The primers used for qRT-PCR analysis are listed in Table S2.

Electron microscopy

The cells were fixed in 4.0% glutaraldehyde in 0.1 M cacodylate buffer for 8 h, postfixed with 1% osmium tetroxide for 1 h, dehydrated in ethanol and acetone (70% ethanol containing 2% uranyl acetate), and embedded in Spi-pon 812 epoxy resin (SPI Inc., Albany, NY, USA). Ultrathin sections were cut using an Ultracut E. Ultratome (Reichert Jung), stained with lead citrate, and photographed using a JEM-1400 electron microscope (Jeol, Tokyo, Japan).

Analysis of the cell cycle

Cells were incubated in the presence of 1 μg/ml EdU for 15 min at 37 °C, washed in PBS, fixed with 3.7% paraformaldehyde for 10 min and permeabilized in 0.5% Triton X-100. EdU was labeled using a Click-iT EdU Cell Proliferation Kit for Imaging, Alexa Fluor 555 dye (Thermo Fisher Scientific, Waltham, MA, USA), according to the manufacturer’s instructions. DNA was stained with 1 μg/ml Hoechst 33342 (Thermo Fisher Scientific, Waltham, MA, USA). Cells were analyzed using a FACSAria SORP cell sorter (BD Biosciences, San Jose, CA, USA). The detection parameters were as follows: Ex. 405 nm, Em. 450/50 BP for Hoechst 33342 and Ex. 561 nm, Em. 585/15 BP for EdU-Alexa Fluor 555.

For Ki-67 staining, cells were fixed with 1% paraformaldehyde for 10 min, washed in PBS, permeabilized in 0.01% Triton X-100 for 10 min, washed in PBS and stained with anti-Ki-67 PE-conjugated antibodies (BD Pharmingen, San Diego, CA, USA) according to the manufacturer’s instructions. Cells were analyzed using a FACSAria SORP cell sorter (BD Biosciences, San Jose, CA, USA). PE fluorescence was detected at Ex. 561 nm, Em. 585/15 BP.

Analysis of apoptosis

Cell death was analyzed by flow cytometry using a FACSAria SORP instrument (BD Biosciences, San Jose, CA, USA). Cells were simultaneously stained with 1 μg/ml Hoechst 33342 (Thermo Fisher Scientific, Waltham, MA, USA), 100 nM TMRE (tetramethylrhodamine, ethyl ester, perchlorate, Thermo Fisher Scientific, Waltham, MA, USA) and annexin V-Alexa Fluor 647 (Thermo Fisher Scientific, Waltham, MA, USA) according to the manufacturer’s instructions. Thus, the DNA content, mitochondrial membrane potential and phosphatidylserine externalization were analyzed together for each sample. The detection parameters were as follows: Ex. 405 nm, Em. 450/50 BP for Hoechst 33342; Ex. 561 nm, Em. 585/15 BP for TMRE and Ex. 647 nm, Em. 640/14 BP for annexin V-Alexa Fluor 647. Additionally, the cells were stained with anti-caspase 3 PE-conjugated antibodies (PE active caspase-3 apoptosis kit; BD Biosciences, San Jose, CA, USA). Cell fixation, permeabilization and staining were performed according to the manufacturer’s instructions. Active caspase 3-PE was detected at Ex. 561 nm, Em. 585/15 BP using a FACSAria SORP cell sorter (BD Biosciences, San Jose, CA, USA).

Chromosome preparations, FISH and cytogenetic analysis

For metaphase chromosome preparations, colcemid (Invitrogen, Waltham, MA, USA) was added to cultivation media at a final concentration of 0.1 μg/ml. Cells were collected 3 h after the addition of colcemid. Hypotonic treatment (0.075 M KCl) was performed for 15 min at 37 °C. Cells were fixed with an ice-cold mix of methanol and glacial acetic acid (3:1). Metaphase slides were made according to standard procedures and used for FISH one day after preparation. FISH was performed according to the manufacturer’s recommendation with a mix of DNA probes specific to whole human chromosomes 1, 2 and 4 (Metasystems, Altlußheim, Germany). The DNA probes to human chromosomes 1, 2 and 4 were labeled by fluorochromes of red, green and both colors, respectively.

Metaphase identification and image acquisition were performed with the slide scanning platform Metafer (v.3.11.8, Metasystems, Altlußheim, Germany) installed on an upright light microscope (Axioscope A1, Carl Zeiss, Germany). For image processing, the Isis FISH imaging system (v5.5, release 5.5.10, Metasystems, Altlußheim, Germany) was used. The number of metaphases scored per sample in each replicate varied from 1,091 to 1,592.

We used Fisher’s exact test to estimate the statistical significance of differences in the level of chromosomal aberrations. Differences were considered statistically significant at a significance level of p < 0.01.

Results

Generation of cell lines for the analysis of HIV-1 Tat action in B cells

To analyze the effect of HIV-1 Tat on B cells, we developed RPMI 8866-based cell lines stably expressing Tat protein fused with EGFP (hereafter referred to as RPMITat) (Fig. 1A). As controls, we constructed cell lines expressing either EGFP (RPMIEGFP) or TatC22G-EGFP, a mutant Tat protein deprived of transactivation capacity (RPMICys).

Figure 1 Cell lines that were used to analyze HIV-1 Tat action of cultured B cells.

(A) Four cell lines that were used in this study. Created with BioRender.com. (B) EGFP fluorescence of demonstrated high purity and homogeneity of the obtained cell lines (cells without EGFP fluorescence are colored blue, and those with EGFP fluorescence are colored green). (C) Western blot analysis of EGFP, Tat-EGFP and TatC22G-EGFP expression in the cell lines. (D) The transactivation ability of Tat-EGFP in HeLa cells with integrated LTR-TurboRFP (flow cytometry, a representative experiment). TurboRFP fluorescence was clearly detected after the expression of Tat-EGFP but not in nontransduced cells (control) or after the expression of EGFP.

Stable lines were obtained by transduction of cells with pseudoviral particles, and cells expressing the proteins of interest were selected using a fluorescence-activated sorter (FACS) (Fig. 1B). Flow cytometry demonstrated high purity and homogeneity of the obtained cell lines. Of note, there was an admixture of nonfluorescent cells in the RPMITat and RPMICys cell lines. Tat-EGFP and TatC22G-EGFP expression was confirmed by western blotting (Fig. 1C). Tat-EGFP and TatC22G-EGFP were partially proteolysed, and as a result, additional bands at ~27 kDa were visible (nonprocessed images are presented in Fig. S1). As the EGFP tag could have interfered with Tat activity, we analyzed the transactivation capacity of Tat-EGFP and TatC22G-EGFP using an in vitro assay based on the fast-maturing fluorescent protein TurboRFP. TurboRFP expression in HeLa cells was controlled by a fragment of the HIV-1 3′ LTR (Kurnaeva et al., 2022). We transduced EGFP, Tat-EGFP, or TatC22G-EGFP into these HeLa cells and found that the expression of Tat-EGFP substantially increased TurboRFP fluorescence compared to EGFP, which did not cause an increase in TurboRFP expression (Fig. 1D). TatC22G had a ~17 fold decreased transactivation activity as compared to Tat (the median fluorescence intensity of TurboRFP in Tat-expressing cells was 34601, the median fluorescence intensity in TatC22G-expressing cells was 2009, and the median fluorescence intensity in EGFP-expressing cells was 309 in the representative experiment shown in Fig. 1D). Hence, the transactivation activity of the Tat protein was not perturbed by its fusion with EGFP.

HIV Tat protein affects the gene expression profile of RPMI 8866 cells

To determine the genes regulated by HIV-1 Tat, total RNA from RPMI 8866, RPMIEGFP, RPMITat and RPMICys cell lines was collected, and RNA-seq was performed. Three biological replicates were sequenced for each cell line. From 16 to 24 million 51 nt sequencing reads were generated for the RPMITat, RPMICys, RPMIEGFP and RPMI samples. On average, 67.17% of reads in every sample were uniquely aligned to the reference genome GRCh38, and 78.93% of them were nonambiguously mapped to the GENCODE gene annotation (Fig. S2). To assess variability between replicates and between cell lines, rlog-transformed filtered count data were visualized by principal component analysis (Fig. S3A). Biological replicates proved to be highly alike by clustering tightly according to sample type. RPMITat samples tended to cluster with RPMICys samples, implying similar effects of Tat-EGFP and TatC22G-EGFP on gene expression. This finding was also confirmed by Spearman correlation of normalized gene expression profiles (Fig. S3B).

To identify changes in gene expression induced by Tat protein in RPMI 8866 cells, we performed differential expression analysis for each cell line. We qualified differentially expressed genes (DEGs) with p.adj < 0.05 and fold change >1.5 in any direction.

Comparison of the control RPMIEGFP against the RPMI cell line demonstrated that the impact of EGFP on gene expression could be neglected, as the analysis revealed only 17 upregulated and 36 downregulated DEGs, implying that the gene expression profiles of RPMIEGFP and RPMI samples were highly similar (Fig. 2A). Therefore, we further used RPMI cells as the control and their gene expression level as the baseline to identify genes regulated by Tat or TatC22G proteins.

Figure 2 Differentially expressed genes (DEGs) in RPMI cells expressing EGFP, EGFP-Tat or EGFP-TatC22G.

(A) The number of all DEGs (left) and protein-coding DEGs (right) found in three comparisons: RPMIEGFP vs RPMI, RPMITat vs RPMI, and RPMICys vs RPMI. (B) Validation of the RNA-seq dataset using qRT-PCR on the indicated upregulated and downregulated genes (mean ± SEM; n = 3). (C) Upregulated (left) and downregulated (right) GO BP terms affected by Tat identified by GSEA (RFPITat cells vs RPMI cells). Only significantly enriched (adjusted p value < 0.05) and nonredundant GO BP terms are shown (the top 20).

Out of 32,120 genes with detectable expression, 1,038 genes were differentially expressed between RPMITat and RPMI cells. A comparable number of DEGs (902 genes) were identified in the comparison RPMICys vs the control (Fig. 2A). Among the detected DEGs, slightly more genes were downregulated (60% or 55%) than upregulated in the presence of Tat or TatC22G protein, respectively. The majority of genes whose expression was affected were protein-coding genes. We validated the expression of several DEGs using quantitative real-time polymerase chain reaction (qRT-PCR) assays, thus confirming the reliability of RNA-seq (Fig. 2B).

RPMI 8866 is an EBV-positive B-lymphoblastoid cell line derived from a patient with chronic myelogenous leukemia (McCune, Fu & Kunkel, 1981). RPMI 8866 cells express EBNA1, -2, -3A, -3B, -3C, and LMP-1, -2A, and -2B proteins and several noncoding RNAs. To explore the possibility that changes in gene expression were a result of Tat-induced changes in the expression of EBV genes, we performed differential gene expression analysis of the EBV transcriptome and found no evidence of Tat or TatC22G protein impact on the expression of viral genes in RPMITat or RPMICys cells (Fig. S4); thus we concluded that we observed a direct effect of Tat or TatC22G on the host cell.

Next, we performed gene set enrichment analysis (GSEA) and overrepresentation analysis (ORA) to search for activated or suppressed functional gene groups and molecular pathways as defined by the GO Biological Process (GO BP) and KEGG databases.

The analysis of modified biological processes (GO BP) was performed by GSEA with subsequent removal of redundant GO terms with REVIGO (Fig. 2C). Comparison of RPMITat cells with the control RPMI cells showed that genes whose expression was upregulated in the presence of Tat protein were enriched for antiviral responses, including Regulation of defense response to virus (p.adjust = 4.3 × 10−2), Type I interferon production (p.adjust = 1.9 × 10−2), Type I interferon signaling pathway (p.adjust = 1.1 × 10−3), Viral genome replication (p.adjust = 1.4 × 10−2), Regulation of viral process (p.adjust = 1.6 × 10−2), and Negative regulation of viral processes (p.adjust = 9.2 × 10−3) (Fig. 2C, left panel). In addition to the activation of pathways associated with antiviral responses, the activation of Regulation of DNA-template transcription, elongation (p.adjust = 5.9 × 10−3), mRNA processing (p.adjust = 2.2 × 10−3), Regulation of mRNA processing (p.adjust = 4.3 × 10−2), and G1/S phase transition of mitotic cell cycle (p.adjust = 2.5 × 10−2) was also observed in RPMITat cells. These changes may be either part of the global antiviral response or the result of the action of a viral protein (proviral reaction). Among the downregulated pathways, pathways associated with cell adhesion (Adherens junction organization (p.adjust = 3.1 × 10−2), Cell-cell adhesion (p.adjust = 2.6 × 10−6), Positive regulation of cell adhesion (p.adjust = 2.5 × 10−2)), and multiple metabolic and biosynthetic pathways were detected (Fig. 2C, right panel). Notably, genes involved in the regulation of cellular proliferation were downregulated (Regulation of leukocyte proliferation (p.adjust = 2.5 × 10−2), Lymphocyte proliferation (p.adjust = 8.4 × 10−3)).

We additionally analyzed changes in signaling pathways, for which overrepresentation analysis (ORA) using KEGG databases was performed (Fig. 3). Analysis of up- and downregulated DEGs revealed activation of innate immunity pathways, such as Toll-like receptor signaling pathways (hsa04620; p.adjust = 3.1 × 10−2) and NOD-like receptor signaling pathways (hsa04621; p.adjust = 3.1 × 10−2), as well as the Influenza A (hsa05164; p.adjust = 3.1 × 10−2) pathway, which was highly overlapping with the former two. Herpes simplex virus 1 infection pathway (hsa05168; p.adjust = 3.1 × 10−2) seems to be related to the antiviral response (Fig. 3A). Several zinc finger proteins (ZNFs) were annotated as related to this pathway. Notably, several genes of the JAK-STAT signaling pathway (JAK1 and STAT1) were general for all detected pathways (Fig. 3B). Secreted interferon-I induces IFN-stimulated genes through the JAK-STAT pathway (Ivashkiv & Donlin, 2014). Some ZNFs regulate the transcription of interferon-stimulated genes (Wang & Zheng, 2021), but among ZNF genes detected by RNA-seq, there was only one well-investigated protein: ZNF268 (Table S3). ZNF268 was described as chronic lymphocytic leukemia (CLL)-associated antigen (Krackhardt et al., 2002), its aberrant alternative splicing was detected in human hematological malignancies (Zhao et al., 2008). ZNF268 also contributes to cervical carcinogenesis through the NF-кB signaling pathway (Wang et al., 2012).

Figure 3 Enrichment analysis of Tat-affected DEGs (RPMITat vs RPMI).

(A) KEGG pathways positively regulated by Tat, identified by ORA of protein-coding DEGs. Only significantly enriched (adjusted p value < 0.05) KEGG pathways are shown. (B) Activated KEGG pathways (hsa04620, hsa04380, hsa04621, and hsa05168) and associated DEGs after filtering overlapping gene sets. (C) KEGG pathways negatively regulated by Tat, identified by overrepresentation analysis of protein-coding DEGs. Only significantly enriched (adjusted p value < 0.05) KEGG pathways are shown. (D) Suppressed KEGG pathways (hsa04514, hsa04060, and hsa04670) and associated DEGs after filtering overlapping gene sets.

Downregulated DEGs were enriched for Cytokine-cytokine receptor interaction pathway (hsa04060; p.adjust = 2.0 × 10−6) and overlapped with Viral protein interaction with cytokine and cytokine receptor pathway (hsa04061; p.adjust = 2.9 × 10−5), reflecting possible inhibition of the proinflammatory response (Fig. 3C). The downregulated genes of these pathways included several cytokines (IL16, IL19, TNFSF9, CCL20, CCL22, CFS1, etc.) and receptors of cytokines (IL1R1, IL6R, IL7R, CCR4, CCR8, TNFRSF19, TNFRSF11A, etc.) (Fig. 3D). At the same time, some proinflammatory cytokines were upregulated (for example, IL6, FC 2.12). Additionally, pathways associated with cell adhesion molecules (hsa04514; p.adjust = 8.0 × 10−7), and Leukocyte transendothelial migration (hsa04670; p.adjust = 2.2 × 10−2) were also suppressed in RPMITat cells, indicating possible inhibition of cell adhesion (Fig. 3C). Among the genes of cell adhesion molecules pathway, several genes (HLA-DQA1, HLA-DQB1, HLA-DRB1, and HLA-DRB5) coding an MHC class II cell surface receptor were suppressed, which might affect the T cell receptor signaling pathway (Fig. 3D).

Additionally, we analyzed the expression of different transcription factors because such changes can lead to pronounced effects. We found that the expression of 12 transcription factors was upregulated (ASCL1, ATF5, FOXO3, NR2F2, PBX1, PRDM5, REL, SIX1, STAT1, TFCP2, ZNF268, ZNF740), and 13 were downregulated (DMRT2, FLI1, HOXB7, JUN, KDM3A, MYCN, NR1H4, SOX5, SOX6, SSX1, ZC3H6, ZNF358, ZNF613); thus, the action of Tat (direct or indirect) may be at least partially connected with the regulation of transcription factor expression.

In addition to differentially expressed protein coding genes, the expression levels of 154 long non-protein coding RNAs (lncRNAs) were altered by HIV-1 Tat expression (54 and 100 lncRNAs were upregulated and downregulated, respectively). Differentially expressed lncRNAs accounted for 14.8% of all DEGs between RPMITat and control cells. These lncRNAs are poorly characterized, but two genes of lncRNAs among the identified DEGs are well known. The MALAT1 lncRNA involved in transcriptional regulation and alternative splicing (West et al., 2014) was upregulated 2.44-fold in RPMITat over the control (the elevated expression was additionally confirmed by qRT-PCR, Fig. S5A). MALAT1 is upregulated in HIV-1-infected CD4+ T cells (Qu et al., 2019) and in the peripheral blood of HIV-1-infected patients (Jin et al., 2016). NEAT1 lncRNA forms the core structural component of paraspeckle bodies and is known for its contribution to HIV replication (Zhang et al., 2013). We observed a 1.66-fold increase in NEAT1 expression in the RPMITat B cell line (see also Fig. S5B), indicating that NEAT1 may play an important role in the development of cell responses induced by the Tat protein.

We also analyzed other DE lncRNAs. The complete list of DE lncRNAs is presented in Table S4, and, since the level of some of these lncRNAs was extremely low, we additionally analyzed ten most expressed DE lncRNAs (Table 1). Some of DE lncRNAs, which were differentially expressed in RPMITat cells, are also differentially expressed in different tumors and might be involved in development of these tumors. The most known examples are MALAT1 and NEAT1 expressed in different neoplasms, including B-cell lymphomas and leukemias (Table S5).

Table 1 Upregulated and downregulated long non-coding RNAs (PPMITat vs RPMI cells) with the highest expression (top 10).

Upregulated genes	
EnsID	Gene name	padj	log2FC	FC	Mean count	Functions and possible involvement in oncogenesis	Reference	
ENSG00000251562	MALAT1	9.9E−22	1.28	2.44	34,783.28	Table S5	
ENSG00000245532	NEAT1	2.2E−08	0.73	1.66	3,039.09	Table S5	
ENSG00000260658	RP11-368L12.1	7.0E−05	0.67	1.59	1,545.91	Co-expressed with gene module of actively proliferating pre-B cells	(Petri et al., 2015)	
ENSG00000248837	RP11-412P11.1	9.5E−08	0.99	1.99	859.73	–		
ENSG00000230448	LINC00276	2.6E−04	0.59	1.50	517.66	–		
ENSG00000234663	LINC01934	1.4E−09	0.90	1.86	516.91	Downregulated in thyroid carcinoma	(Zhang et al., 2019)	
ENSG00000226965	AC003088.1	2.3E−11	1.19	2.29	319.39	Upregulated in breast cancer line	(Li et al., 2021)	
ENSG00000238129	RP3-410C9.2	8.2E−15	0.86	1.82	307.45	–		
ENSG00000235385	LINC02154	8.5E−79	2.43	5.40	272.56	Upregulated in Laryngeal Squamous Cell Carcinoma and renal cell carcinoma, and can be used as a prognostic feature; promotes the proliferation and metastasis of hepatocellular carcinoma	(Zuo et al., 2018; Gong et al., 2020; Yue et al., 2022)	
ENSG00000249645	RP11-552M14.1	2.0E−22	1.19	2.28	261.05	–		
Downregulated genes	
ENSG00000253377	RP11-566H8.3	6.4E−07	−0.86	0.55	1,751.51	Produces non-canonical cancer/testis antigen	(Chong et al., 2020)	
ENSG00000254119	RP11-705O24.1	9.9E−16	−1.40	0.38	1,623.69	Potential prognostic feature in esophageal cancer	(Fan & Liu, 2016)	
ENSG00000251381	LINC00958	9.4E−23	−0.68	0.62	1,003.09	Canonical lncRNA in human cancer progression (overexpressed in many cancers)	(Li et al., 2022)	
ENSG00000267761	CTD-2130O13.1	4.5E−08	−1.13	0.46	895.69	Potential cancer-specific biomarker	(Mohammed et al., 2017)	
ENSG00000271955	RP11-444A22.1	3.4E−79	−3.03	0.12	803.23	Involved in acute myeloid leukemia cell differentiation	(Cozzi et al., 2022)	
ENSG00000231772	RP1-154K9.2	8.3E−53	−9.37	0.00	726.17	–		
ENSG00000227681	RP11-307P5.1	4.5E−10	−1.17	0.44	621.53	–		
ENSG00000251088	RP11-325B23.2	1.5E−04	−0.63	0.65	454.27	–		
ENSG00000185904	LINC00839	2.5E−62	−1.88	0.27	332.95	Downregulated in gastric cancer with H. pilori infection; upregulated in chemoresistant breast cancer	(Zhong et al., 2018; Chen et al., 2020)	
ENSG00000242741	LINC02005	1.6E−20	−1.44	0.37	324.00	–		
Note:

The full list of DE lncRNAs is presented in Table S4.

Comparison of Tat- and TatC22G-induced effects

HIV-1 Tat can modify host gene expression by several nonrelated mechanisms. Tat can transactivate host genes by binding to TAR-like sequences in their nascent mRNAs, e.g., IL6 mRNA (Ambrosino et al., 1997) or TNF-β mRNA (Buonaguro et al., 1994). Tat can also associate with chromatin and control RNA polymerase II recruitment and pause release (Reeder et al., 2015). Additionally, HIV-1 Tat interacts with hundreds of nuclear proteins (Gautier et al., 2009), and this interaction may also modify the expression of host genes.

Transactivation of transcription by the mutant TatC22G protein was significantly weaker than that by Tat (Fig. 1D), but it could still bind to chromatin and interact with other proteins. Both Tat and TatC22G proteins modulate the expression of similar and overlapping sets of genes (Fig. 4A). Comparison of the transcriptome profiles of the RPMITat and RPMICys cell lines revealed 201 protein-coding DEGs (p.adj < 0.05; fold change >1.5 in any direction).

Figure 4 Comparison of gene expression between RPMITat and RPMICys cells.

(A) Venn diagrams demonstrated that approximately half of the DEGs overlapped, indicating similar but not identical modifications induced by EGFP-Tat and its mutated form EGFP-TatC22G. (B) Suppressed KEGG pathways (hsa04060, hsa04080, hsa05033, and hsa05032) and associated DEGs when comparing RPMICys against RPMITat cells.

Overrepresentation analysis against the KEGG database also demonstrated little gene set enrichment, reflecting relatively small differences between RPMICys and RPMITat cells (Fig. 4B). Notably, the Cytokine-cytokine receptor interaction pathway (hsa04060) downregulated by HIV-1 Tat was further suppressed by mutant TatC22G protein in RPMICys cells compared to RPMITat cells.

The pathological effects of Tat expression

Our RNA-seq analysis identified several pathways affected by Tat. We next experimentally analyzed key characteristics of RPMI 8866, RPMIEGFP, RPMITat, and RPMICys cell lines to identify how Tat affected B cells.

Electron microscopy demonstrated that the expression of EGFP, Tat-EGFP or TatC22G-EGFP did not lead to any substantial changes in cellular organization (Fig. 5A). To ascertain whether changes in the expression of several genes involved in the G1/S phase transition and cell proliferation (Fig. 2C) affected cell cycle progression in Tat-expressing cells, we analyzed the proportion of G1/G0, S, and G2/M cells in the total population of RPMITat cells using flow cytometry. We could not detect any significant changes in the proportions of cells at different stages of the cell cycle using this approach (Figs. 5B, 5C). We also detected S-phase cells using incorporation of EdU and did not observe any difference between different cell lines (Fig. 5D). Finally, we analyzed fractions of cycling (non-G0) cells using antibodies against the proliferation marker Ki-67, and again, no difference was found (Fig. 5E).

Figure 5 EGFP, Tat-EGFP or TatC22G-EGFP expression does not affect the morphology or proliferative potential of RPMI 8866 cells.

(A) Representative cells of RPMI, RPMIEGFP, RPMITat, and RPMICys lines under electron microscopy. Bars = 1 μm. (B) Cell cycle distribution of RPMI, RPMIEGFP, RPMITat, and RPMICys cells (representative experiment) and (C) estimation of cell proportions at the G0/G1, S, and G2/M stages (mean ± SD, n = 3). (D) Estimation of S-phase cells using incorporation of synthetic nucleotides (EdU) (mean ± SD, n = 3). (E) Estimation of cycling cells (Ki-67-positive) (mean ± SD, n = 3). (F) Estimation of the content of apoptotic cells in RPMI, RPMIEGFP, RPMITat, and RPMICys lines using four independent methods (mean ± SD, n = 3). In C–F, all differences between control cells (RPMI) and cells expressing different proteins (RPMIEGFP, RPMITat, and RPMICys) were insignificant (Kruskal–Wallis test, p > 0.05; n = 3).

Cell death can also influence the dynamics of cell populations; therefore we analyzed apoptosis in RPMITat cells. We did not observe any statistically significant changes in the content of sub-G1 cells (dead cells), cells with reduced mitochondrial potential (TMRE staining, which allowed for marker cells during very early stages of apoptosis), proportion of annexin V-positive, or caspase 3-positive cells (markers of apoptotic cells) (Fig. 5F).

Thus, stable expression of Tat did not produce strong effects that could be detected using standard analysis of cellular proliferation and apoptosis. At the same time, subtle, but persistent changes (if they exist) could potentially influence the dynamics of cell growth. Therefore, we cultured cells for three months and found that the percentage of cells expressing Tat-EGFP was gradually reduced (Figs. 6A, 6B). Such replacement of Tat-expressing cells by non-expressing cells was not detected in cells expressing either TatC22G-EGFP or EGFP, clearly indicating that this effect was a consequence of Tat protein per se.

Figure 6 HIV-1 Tat influences cellular dynamics and chromosome organization upon prolonged cultivation.

(A) Phase contrast images and EGFP fluorescence (merged) in RPMI, RPMIEGFP, RPMITat, and RPMICys after 3 months of cultivation (representative images). In the RPMITat line, there are ~40% cells without EGFP fluorescence (arrowheads). (B) The proportion of EGFP-positive cells was decreased during prolonged cultivation of RPMITat but not RPMIEGFP and RPMICys cells (a representative experiment). (C) Representative image of a metaphase plate with detected chromosomes 1 (red), 2 (green) and 4 (orange). The chromosome with a translocation is marked with an arrow. (D) Cells expressing Tat-EGFP or TatC22G-EGFP contained a significantly higher proportion of chromosomes with aberrations (see Table S6 for a detailed description of chromosome aberrations). The comparison was performed using Fisher’s exact test. An asterisk (*) indicates differences were considered statistically significant at p < 0.01.

To detect possible mutagenic effects of HIV-1 Tat expression, we performed cytogenetic analysis. We scored chromosome aberrations involving chromosomes 1, 2 and 4 revealed by fluorescence in situ hybridization with whole chromosome paints (Fig. 6C). The images of metaphases were automatically acquired, and then metaphases were analyzed for the presence of structural chromosomal aberrations, including translocations, dicentrics, acentrics, and deletions. Numerical chromosomal abnormalities, such as polyploidy and aneuploidy, were also taken into consideration. We observed a reproducible increase in the frequency of translocations as well as in the total yield of structural aberrations in cells expressing either Tat-EGFP or TatC22G-EGFP (Fig. 6D; Table S6).

Discussion

To identify the mechanisms potentially leading to the development of B-cell lymphomas in HIV-infected patients, we investigated the effect of Tat protein on gene expression in cultured B cells. Upon long-term exposure of B-cells to Tat, the effects could accumulate and potentially provoke an oncogenic transformation. Therefore, we did not use either an ex vivo experimental model, in which lymphocytes isolated from the blood of healthy donors were incubated in the presence of Tat protein (for example, (Germini et al., 2017)), or a model with the expression of HIV-1 controlled by an inducible promoter (for example, (Reeder et al., 2015)). These experimental systems are more likely to simulate an acute infection situation, but they do not allow for the study of weak long-term effects of protein and compensatory reactions of the cell. In the current study, we obtained the B lymphoid cell line RPMI 8866 stably expressing HIV-1 Tat fused with EGFP and used these cells to simulate a prolonged systemic effect of the presence of Tat in blood of chronically infected patients.

To detect possible changes provoked by chronic exposure of B-cells to Tat, we performed a genome-wide analysis of cellular gene expression (RNA-seq). Tat protein induces differential expression of approximately 1,000 genes (p.adj < 0.05 and fold change of ≥1.5 in any direction). To predict the effects of these changes, the analysis of metabolic and signaling pathways (GO BP and KEGG analysis) was performed. GO BP analysis demonstrated the activation of pathways involved in cellular antiviral reactions, and suppression of different metabolic pathways and proliferation. KEGG analysis also demonstrated activation of innate immunity pathways involved in antiviral reactions: toll-like receptor signaling and NOD-like receptor signaling pathways. We observed an increase in JAK1 and STAT1 expression, indicating that Tat may affect these pathways via the JAK-STAT signaling. Additionally, we found the reduction of Cytokine-cytokine receptor interaction pathway. Downregulation of this pathway (and downregulation of metabolic pathways and proliferation) may result from direct HIV-1 Tat action (proviral reactions). It should be noted however that not all cytokines and their receptors were downregulated. HIV-1 Tat induces the expression of several proinflammatory cytokines, mainly IL-6 in several cell types (Ambrosino et al., 1997; Nookala & Kumar, 2014; Ben Haij et al., 2015). Our results also confirm the upregulation of pro-inflammatory IL6 gene in Tat-expressing B cells. Thus, the gene expression pattern we observed in Tat-expressing B cells probably resulted from a combination of Tat action per se and cellular antiviral reactions.

We also analyzed the lncRNA expression. Differentially expressed lncRNAs accounted for 14.8% of all DEGs between RPMITat and the control cells. While functions of the majority of identified lncRNA are poorly studied, some of them (e.g., NEAT1 and MALAT1) play a role in development of B-cell neoplasms (Table S5). In Tat-expressing RPMI 8866 cells, both of these lncRNAs were upregulated. It is possible that these or some other DE lncRNAs can be involved in the development of B-cell lymphomas in HIV-1-infected patients, but this requires further study.

Another important observation on the mode of action of Tat in host cells came from the comparison of the action of Tat and its TatC22G mutant with the reduced transactivation activity. In our experiments, 313 protein-coding genes were regulated by Tat only (115 upregulated and 198 downregulated, respectively), 228 genes were regulated by TatC22G only (117 upregulated and 111 downregulated genes), while the most genes (435) were regulated both by Tat and TatC22G (185 upregulated and 250 downregulated genes) (Fig. 4). HIV-1 Tat can modify cellular processes via different mechanisms. It can transactivate host genes by binding to TAR-like sequences (for example, this mechanism seems to regulate the transcription of IL6 in T cells (Ambrosino et al., 1997) and TNF-b in B cells (Buonaguro et al., 1994)). Tat can also interact with chromatin and RNA-polymerase II (Reeder et al., 2015) as well as with hundreds of other cellular and nuclear proteins (Gautier et al., 2009). Therefore, to distinguish between the two potential modes of action of Tat, we compared the effects of HIV-1 Tat and that with the C22G mutation possessing a significantly decreased transactivation activity. We hypothesized that genes differentially expressed between RPMITat and RPMICys cells would be associated with the transactivator activity of this protein while genes similarly regulated would rather be affected by protein-protein interactions involving Tat. The RNA-seq results demonstrated that most of the effects of EGFP-Tat and EGFP-TatC22G were similar, and hence, the main mechanism of prolonged Tat action on B-cells seems to be due to Tat interaction with host proteins.

We next analyzed some physiological parameters of RPMITat cells. We could not find any manifested changes in cellular morphology, proliferation, or apoptosis. At the same time, we observed an effect of prolonged cultivation, i.e., a decrease in EGFP-Tat-expressing cells, indicating that Tat inhibited cell growth. Thus, although there were no strong effects of EGFP-Tat expression, a weak effect, which manifested itself only in situations of long observation, was induced. Additionally, we found that chromosome aberrations occurred more frequently in cell lines expressing EGFP-Tat and EGFP-TatC22G. These observations are consistent with published data obtained using an ex vivo model (El-Amine et al., 2018). Modification of cell cycle progression and chromosome aberrations may both promote lymphomagenesis.

Thus, the presence of HIV-1 Tat can in the long run modify the cellular physiology and genome stability of cultured B cells and, as a result, may promote oncogenic transformation. The precise mechanisms of these effects will be the subject of our future work.

Supplemental Information

Supplemental Information 1 Supplementary figures.

Click here for additional data file.

Supplemental Information 2 Differential expression of host genes in RPMI8866 cells ectopically expressing either EGFP, Tat-EGFP or TATC22G-EGFP.

Click here for additional data file.

Supplemental Information 3 Primers used in this study.

Click here for additional data file.

Supplemental Information 4 Differentially expressed zinc finger proteins (extracted from UniprotKB database).

Click here for additional data file.

Supplemental Information 5 Upregulated and downregulated long non-coding RNAs (PPMI Tat vs RPMI cells).

Click here for additional data file.

Supplemental Information 6 Differential expression of MALAT1 and NEAT1 lncRNAs in B-cell-derived neoplasms (Lnc2cancer 3.0 database, http://bio-bigdata.hrbmu.edu.cn/lnc2cancer/).

Click here for additional data file.

Supplemental Information 7 Chromosomal aberrations in RPMI8866 cells ectopically expressing either EGFP, Tat-EGFP or TatC22G-EGFP.

Click here for additional data file.

We are grateful to A.V. Lazarev for technical support. The following reagent was obtained through the NIH AIDS Reagent Program, Division of AIDS, NIAID, NIH: HIV-1 HXB2 GST-Tat Expression Vector (GST-Tat 1 (86R)) from Dr. Andrew Rice. This work was carried out within the framework of the International Research Network (IRN) ONCO3D and the IDB RAS Government basic research program (0088-2021-0007). The flow cytometry facility became available in the framework of the Moscow State University Development Program.

Additional Information and Declarations

Competing Interests

Author Contributions

DNA Deposition

Data Availability

Yegor S. Vassetzky is an Academic Editor for PeerJ.

Anna A. Valyaeva performed the experiments, analyzed the data, prepared figures and/or tables, authored or reviewed drafts of the article, and approved the final draft.

Maria A. Tikhomirova performed the experiments, analyzed the data, prepared figures and/or tables, authored or reviewed drafts of the article, and approved the final draft.

Daria M. Potashnikova performed the experiments, prepared figures and/or tables, and approved the final draft.

Alexandra N. Bogomazova performed the experiments, prepared figures and/or tables, authored or reviewed drafts of the article, and approved the final draft.

Galina P. Snigiryova performed the experiments, prepared figures and/or tables, resources contribution, and approved the final draft.

Aleksey A. Penin performed the experiments, analyzed the data, prepared figures and/or tables, and approved the final draft.

Maria D. Logacheva performed the experiments, prepared figures and/or tables, and approved the final draft.

Eugene A. Arifulin performed the experiments, prepared figures and/or tables, and approved the final draft.

Anna A. Shmakova performed the experiments, analyzed the data, prepared figures and/or tables, authored or reviewed drafts of the article, and approved the final draft.

Diego Germini performed the experiments, prepared figures and/or tables, and approved the final draft.

Anastasia I. Kachalova performed the experiments, prepared figures and/or tables, and approved the final draft.

Aleena A. Saidova performed the experiments, prepared figures and/or tables, and approved the final draft.

Anastasia A. Zharikova performed the experiments, analyzed the data, prepared figures and/or tables, and approved the final draft.

Yana R. Musinova performed the experiments, prepared figures and/or tables, and approved the final draft.

Andrey A. Mironov conceived and designed the experiments, analyzed the data, prepared figures and/or tables, authored or reviewed drafts of the article, and approved the final draft.

Yegor S. Vassetzky conceived and designed the experiments, analyzed the data, prepared figures and/or tables, authored or reviewed drafts of the article, and approved the final draft.

Eugene V. Sheval conceived and designed the experiments, performed the experiments, analyzed the data, prepared figures and/or tables, authored or reviewed drafts of the article, and approved the final draft.

The following information was supplied regarding the deposition of DNA sequences:

GSE182538

The following information was supplied regarding data availability:

Original blot pictures are available in the Supplemental Files.

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
