# Peer review of "Ectopic expression of HIV-1 Tat modifies gene expression in cultured B cells: implications for the development of B-cell lymphomas in HIV-1-infected patients"

_PeerJ, doi:10.7717/peerj.13986_

## Round 0.1 · original submission · Major Revisions

Dear author,

Please see the comments from the reviewers. As suggested, your manuscript requires major revisions before any further consideration.
Best regards,

Ravi

Reviewer 1 ·

Basic reporting

The language used in this article is clear and well articulated except in very few instances where I have made suggestions to improve.
The introduction appropriately presents the updated background in this field (oncogenic role of HIV proteins in lymphoma), and makes reference to important and relevant published work in the field (including, but not limited to, their own previous publications).
The overall format of the article is acceptable and conforms to that suggested for publication in PeerJ, and collectively, the results present "self-contained" body of work.

Experimental design

The research question is clear, the experimental design appropriate, and the findings presented in a clear and coherent manner. The findings add new data to the existing body of work on the role of HIV-1 Tat in the development of B cell lymphoma. The methodologies are presented clearly and in detail, the experiments have been performed to a high standard, and the results analyzed and presented in a clear and statistically sound way. There are however some areas which are unclear, and require attention by the author. I have provided an annotated version of the manuscript with my comments.

Validity of the findings

As mentioned earlier, the findings of this study adds new data with regards to the oncogenic role of HIV-1 Tat in the development of B cell lymphoma. There are areas however which need attention, in order to strengthen the impact of the manuscript even further. The authors should refer to my annotated comments on the PdF version of the manuscript which I have attached.

Additional comments

No other additional comments.

Annotated reviews are not available for download in order to protect the identity of reviewers who chose to remain anonymous.

Reviewer 2 ·

Basic reporting

This is a manuscript from the laboratories of Yegor Vassetzky and Eugene Sheval who analyzed the effect of HIV-1 Tat expression in B cells in order to understand the mechanism of HIV-1-assocaited B cell lymphomas. The authors generated B cell lines expressing WT Tat-EGFP and
Transcription activation deficient Tat C22G mutant -EGFP and conducted RNA Seq analysis to identify differentially expressed genes.

Experimental design

1. Transcription activity of Tat-EGFP needs to be tested along with the Tat without EGFP to demonstrate its transactivation capacity. Fig.1D shows only apparent ~2-fold difference between WT Tat-EGFP and Tat C22G -EGFP which indicate poor activation activity of the Tat-EGFP. I suggest using standard Tat-transactivation assay to show that Tat-EGFP can induce basal transcription by 20-40 fold and that Tat C22G-EGFP is deficient in transactivation.
2. Figure 1 legend describes the results, it should only state what is on the figure and the results should be describe in the text. Please correct, and also follow the same logic for other figures where this also happens.
3. Fig.1 C needs quantification.
4. Fig. 1 D needs quantification. Also, basal transcription needs to be quantified and the transactivation folds need to be shown (comment 1 above). You have description in the text, but it is not sufficient, and not clear if it was repeat/statistically verified.
5. Lane 259, “Stable lines were obtained by transduction of cells with pseudoviral particles”, you need to describe how the cell lines were produced. It is not described in the Methods.
6. It seems that Tat expression reduces inflammation response of B cells (reduction of IL1 and Il6 receptors), so how this explains the effect of inflammation on the development of B cell lymphomas?
7. Fig. 2A, move results discussion to the text.
8. Fig.2C, unclear which results are shown, is this comparison of Tat expression cells to parental of EGFP expressing parental cells?
9. Did you compare Tat versus TatC22G cells by DESeq2? Somehow this analysis seems to be omitted. You only showed comparisons of Tat versus cells and Tat C22G versus cells in Fig. 4.

Validity of the findings

Tat expression correlated with the increased expression of number of transcription factors and upregulation of anti viral and interferon responses. Curiously, Tat expression led to the reduced expression of ILR1 and IL-6 receptors indicated reduced inflammatory response of the transduced cells. The authors found no evidence of the Tat expression on B cell proliferation. Moreover, long term culture showed that Tat expressing cells proliferated less efficiently that the non-transduced cells. The authors confirmed that Tat expression led to chromosomal aberrations as previously shown. The main conclusion of the authors, that long-term Tat expression can lead to the increased oncogenicity is not supported by this study. Contrary to what they concluded, it seems that ectopic Tat expression has little or nothing to do with the B cells proliferation. So most likely, there need to be another “culprit” such as Nef, for example, that can also be secreted and changes target cells.

Additional comments

10. Somehow you are completely missing the point that Tat induces transcription by competing with the TAR-like sequence in 7SK large complex and recruiting CDK9/cyclin T1. I suggest to add it to your discussion as this is the main mechanism of transcription activation by Tat.

Minor corrections:
1. Lane 61, “One of the most intriguing effects of HIV-1 infection” – should be “One of the most intriguing effect of HIV-1 infection”
2. Lane 72, “HIV-1 produces a small nuclear transcriptional activator protein known as transactivator of transcription (Tat) which regulates viral transcription and simultaneously modulates cellular processes by interacting with different cellular structures, particularly nuclear components (Musinova et al., 2016; Ali et al., 2021).” – need a better statement. Tat binds to TAR RNA and recruits P-TEFb. This is its main function. Please correct and site a proper review paper.
3. Lane 110, “HeLa cells with integrated LTR-TurboRFP” – please describe how these cells were produced and whether they are available for scientific community.
4. Lanes 173-175, change text color to black.
5. Fig.1A legend, do not need to mention the sources of the cell lines as already mentioned in the Methods.
6. Lane 254, “HIV-1 Tat action on B cells” better “HIV-1 Tat action in B cells”
7. Lane 309, “thus we observed a direct effect of Tat” – better” thus we concluded that this was the direct effect of Tat”
8. Lanes 445 – 454, please reduce the font size.

·

Basic reporting

The manuscript fulfills all the basic reporting criteria. A few typos and one format issue need corrections

Experimental design

The study is well designed and is appropriate with a few limitations. As HIV-1 poorly infects B cells, the Tat effects on B cells must be driven by extracellular Tat. However, the manuscript reports the effects on cellular DNA and RNA of a Tat stably expressed by a lymphoblastoid B cell line. According to the authors, this was done to allow observation of a long-term (chronic) exposure to the viral protein. Have the authors considered coculturing Tat expressing B cells with a non-transduced counterpart? By doing so it should be possible to measure the effect of the Tat released by the Tat expressing B cells on non-transduced B cells. Along the same vein: have the authors measured the Tat protein released? If present, how did the authors rule out the contribution of extracellular Tat to the observed effects? Concerning the progressive expansion of B cells not expressing Tat, was the Tat gene silenced or lost?

Validity of the findings

With the caveat mentioned above, the data are robust, statistically sound (but this reviewer is not an expert in bioinformatics) and well-controlled.

---

## Round 0.2 · accepted · Accept

Dear Dr. Vassetzky,

Thank you for your submission to PeerJ.

I am writing to inform you that your manuscript - Ectopic expression of HIV-1 Tat modifies gene expression in cultured B cells: Implications for the development of B-cell lymphomas in HIV-1-infected patients - has been Accepted for publication. Congratulations!

This is an editorial acceptance; publication is dependent on authors meeting all journal policies and guidelines.

Next steps: Your article is being checked and you will receive a list of production tasks shortly. After you complete these tasks, your proofing PDF will be created (please do not proof check your reviewing PDF!).

Reviewer 1 ·

Basic reporting

The concerns and comments submitted by me during the first round of review were satisfactorily addressed by the authors.

Experimental design

Same comment as above.

Validity of the findings

Same comment as above

Additional comments

I have carefully assessed the responses and adjustments made to the manuscript, in response to my review of the manuscript. The authors have duly addressed these.

Reviewer 2 ·

Basic reporting

No concerns

Experimental design

No concerns

Validity of the findings

No concerns

Additional comments

No additional commnets